# Chronic HIV Infection Increases Monocyte NLRP3 Inflammasome-Dependent IL-1α and IL-1β Release

**DOI:** 10.3390/ijms25137141

**Published:** 2024-06-28

**Authors:** Hedda Hoel, Tuva Børresdatter Dahl, Kuan Yang, Linda Gail Skeie, Annika Elisabet Michelsen, Thor Ueland, Jan Kristian Damås, Anne Ma Dyrhol-Riise, Børre Fevang, Arne Yndestad, Pål Aukrust, Marius Trøseid, Øystein Sandanger

**Affiliations:** 1Research Institute of Internal Medicine, Oslo University Hospital Rikshospitalet, 0372 Oslo, Norway; hbhoe@lds.no (H.H.); t.b.dahl@medisin.uio.no (T.B.D.); kuan.yang.s@gmail.com (K.Y.); anemic@ous-hf.no (A.E.M.); thor.ueland@medisin.uio.no (T.U.); borre.fevang@medisin.uio.no (B.F.); arne.yndestad@gmail.com (A.Y.); paukrust@ous-hf.no (P.A.); marius.troseid@medisin.uio.no (M.T.); 2Department of Internal Medicine, Lovisenberg Diaconal Hospital, 0440 Oslo, Norway; 3Department of Infectious Diseases, Oslo University Hospital, Ullevål, 0450 Oslo, Norway; liskei@ous-hf.no (L.G.S.); anriis@ous-hf.no (A.M.D.-R.); 4Faculty of Medicine, Institute of Clinical Medicine, University of Oslo, 0372 Oslo, Norway; 5Department of Clinical and Molecular Medicine, Norwegian University of Science and Technology, 7034 Trondheim, Norway; jan.k.damas@ntnu.no; 6Department of Infectious Diseases, St. Olavs Hospital, 7030 Trondheim, Norway; 7Section of Clinical Immunology and Infectious Diseases, Department of Rheumatology, Dermatology and Infectious Diseases, Oslo University Hospital, Rikshospitalet, 0372 Oslo, Norway; 8Section of Dermatology, Department of Rheumatology, Dermatology and Infectious Diseases, Oslo University Hospital, Rikshospitalet, 0372 Oslo, Norway

**Keywords:** HIV, inflammasome, monocytes, NLRP3, IFI16, IL-1α, IL-1β, TNF

## Abstract

Antiretroviral treatment (ART) has converted HIV from a lethal disease to a chronic condition, yet co-morbidities persist. Incomplete immune recovery and chronic immune activation, especially in the gut mucosa, contribute to these complications. Inflammasomes, multi-protein complexes activated by innate immune receptors, appear to play a role in these inflammatory responses. In particular, preliminary data indicate the involvement of IFI16 and NLRP3 inflammasomes in chronic HIV infection. This study explores inflammasome function in monocytes from people with HIV (PWH); 22 ART-treated with suppressed viremia and 17 untreated PWH were compared to 33 HIV-negative donors. Monocytes were primed with LPS and inflammasomes activated with ATP in vitro. IFI16 and NLRP3 mRNA expression were examined in a subset of donors. IFI16 and NLRP3 expression in unstimulated monocytes correlated negatively with CD4 T cell counts in untreated PWH. For IFI16, there was also a positive correlation with viral load. Monocytes from untreated PWH exhibit increased release of IL-1α, IL-1β, and TNF compared to treated PWH and HIV-negative donors. However, circulating monocytes in PWH are not pre-primed for inflammasome activation in vivo. The findings suggest a link between IFI16, NLRP3, and HIV progression, emphasizing their potential role in comorbidities such as cardiovascular disease. The study provides insights into inflammasome regulation in HIV pathogenesis and its implications for therapeutic interventions.

## 1. Introduction

Efficient antiretroviral treatment (ART) has transformed HIV from a lethal disease to a chronic condition. HIV is, however, associated with significant co-morbidities such as non-AIDS defining cancer and cardiovascular disease (CVD), partly related to incomplete immune recovery and persistent inflammation [1]. The initial destruction of the immune system and in particular CD4 T cells, which is a hallmark of untreated progressive HIV-infection, occurs to a large degree in the Gut-Associated Lymphoid Tissues [2,3,4]. This causes an impaired mucosal barrier, exposing the systemic circulation and subsequently other organs to pathogen-associated molecular patterns (PAMPs) (e.g., lipopolysaccharide [LPS], flagellin and lipotechoic acid [LTA]), and metabolites (e.g., trimethylamine-N-oxide [TMAO]) from the gut microbiota, thereby contributing to a state of chronic immune activation and inflammation [5,6,7,8,9]. Furthermore, co-infection with cytomegalovirus (CMV) or Epstein–Barr virus (EBV) frequently occurs and contributes to the chronic low-grade inflammation in PWH [10,11].

Inflammasomes are multiprotein complexes that can be activated by one of several cytosolic innate immune receptors. The activated receptor binds an adaptor protein termed ASC, which in turn binds to caspase-1 that subsequently cleaves and activates the potent inflammatory cytokine pro-interleukin (IL)-1β and the pore-forming protein gasdermin D, resulting in a massive release of IL-1β and other cytosolic content into the extracellular space [12,13]. In most cell types, inflammasome activation requires two steps. In the first “priming” step, the pro-inflammatory transcription factor NF-κB is activated by innate immune receptors sensing microbial components like LPS or cell damage, or by pro-inflammatory cytokines from nearby cells. This leads to the transcription of both inflammasome receptors, such as NLRP3, and inactive pro-IL-1β. In the second step, the inflammasome receptors sense one form or another of danger molecules, indicating cellular damage, such as extracellular adenosine triphosphate (ATP) or mono sodium urate crystals. Some receptors, such as IFI16, recognize cytosolic microbial DNA [14]. IFI16 may also be crucial for suppressing reactivation of latent EBV [15]. NLRP3 is a special inflammasome-forming receptor because it senses nearby non-apoptotic cell death indirectly through dire changes of the cytosol induced by extracellular damage associated molecular patterns (DAMPs) acting on surface receptors [16]. Recently, CMV was shown to activate NLRP3 inflammasomes in the THP1 monocytes cell line [17]. 

Preliminary data suggest the involvement of IFI16 and NLRP3 inflammasomes in chronic HIV infection. IFI16 has been reported to be required for the death of lymphoid CD4 T cells infected with HIV [18]. Inflammasome components seem to be upregulated in monocytes in PWH, as immunological responders (IR) had lower NLRP3 and caspase-1 levels, but not IFI16 levels, compared to immunological non-responders (INR) [19]. However, whether chronic HIV infection alters the release of the prototypical NLRP3 product, IL-1β from monocytes remains to be investigated. Moreover, whereas most literature has focused on IL-1β, IL-1α related to HIV has gained less attention. IL-1α is synthesized in an active form, is not a substrate for caspase-1 and its regulation and release is more complex [20]. However, the inflammasome-induced gasdermin D pores in the plasma membrane promote maturation and release of IL-1α, making this cytokine both more potent and abundant in the extracellular space [21]. Hence, both IL-1α and IL-1β depend on inflammasome activation to effectively mediate their pro-inflammatory signaling. 

The ability to release the two IL-1 cytokines is of potential relevance for HIV-related comorbidities. We have previously shown that soluble markers of IL-1 activation predict first-time myocardial infarction, independent of HIV-related and traditional risk factors [22], as well as faster lung function decline independent of smoking [23]. In the present study, we investigated whether IFI16 and NLRP3 expression in human monocytes were associated with HIV viremia and CD4 T cell depletion. We also explored NLRP3 inflammasome function in monocytes from ART-treated and untreated PWH, quantifying both IL-1α and IL-1β secretion. Finally, we investigated inflammasome priming, hypothesizing that monocytes from PWH are pre-primed for inflammasome activation upon ATP stimulation.

## 2. Results

### 2.1. PWH and HIV-Negative Donors

PWH were on average 47 years of age, predominantly male. Treated PWH had received ART for a median of 12 years, all were virally suppressed (<50 copies/mL plasma) and had a median CD4^+^ T cell count of 628 cells/µL, whereas untreated PWH had a median CD4^+^ T cell count of 410 cells/µL. The patient characteristics are shown in (Table 1).

### 2.2. IFI16 and NLRP3 Expression Are Negatively Correlated with CD4 T Cell Counts in PWH

We first measured IFI16 and NLRP3 mRNA expression in monocyte from 17 untreated PWH. IFI16 expression, but not NLRP3, correlated positively with viremia, mostly driven by individuals with high viral load (Figure 1A,B). Furthermore, both IFI16 and NLRP3 correlated negatively with CD4 T cell counts, indicating decreased levels in relation to HIV-related immunodeficiency (Figure 1C,D). When stratified into three groups with high, medium, or low viral load (Figure 1E), IFI16 mRNA expression was significantly higher in the group with high viral load compared to medium or low viral load, as well as HIV-negative donors (Figure 1F). A more complex pattern was seen for NLRP3 mRNA expression. Thus, although NLRP3 mRNA expression was seemingly higher in PWH with high compared to low viral load, NLRP3 expression was significantly reduced in monocytes from PWH compared to HIV-negative donors (Figure 1G). Altogether, whereas the associations with viral load were more complex and not significant for NLRP3, our data support that HIV infection induces IFI16 in human monocytes, and both IFI16 and NLRP3 expression is higher in PWH with more pronounced immunodeficiency (i.e., with lower CD4 T cell counts). 

### 2.3. Monocytes from Untreated PWH Secrete More IL-1α, IL-1β, and TNF than Treated PWH

We next investigated the release of IL-1α and IL-1β as well as TNF, as a cytokine that will be released by “signal 1” alone, from monocytes, first primed with medium only or LPS as signal 1 for 1–6 h, and subsequently activated with extracellular ATP as a signal 2 for 30 min (Figure 2A–C). Cytokine secretion from HIV-negative donors cells and cells from ART-treated PWH were of similar levels, although a slightly reduced IL-1α secretion was observed in monocytes from treated PWH. However, monocytes from untreated PWH secreted more IL-1α and IL-1β than both treated PWH and HIV-negative donors. These data further support that chronic HIV infection may increase inflammasome activity in monocytes and that this release at least partly could be attenuated by ART. Furthermore, monocytes from untreated PWH also secreted more TNF, an NLRP3-independent cytokine, in response to LPS. This finding suggests that the priming step may be more efficient in monocytes from untreated PWH compared to monocytes from treated PWH and HIV-negative donors.

We further investigated whether these differences were due to cytokine priming or alternatively caused by an increase in inflammasome components. RT-PCR quantification revealed significantly increased IL-1β mRNA in LPS-stimulated monocytes from untreated PWH compared to that of monocytes from treated PWLH (Figure 3B). IL-1β mRNA levels in untreated PWH were also higher than that of healthy control cells. Monocytes from both treated and untreated PWH had an early and higher increase of TNF mRNA after LPS stimulation than monocytes from HIV-negative donors (Figure 3C), in line with the later increase of TNF protein in untreated PWH (Figure 2C). There were no statistically significant differences in IL-1α, NLRP3, or IFI16 mRNA expression between the groups (Figure 3A,D,E). These data again support that increased cytokine secretion is mediated through more efficient cytokine synthesis rather than increased levels of inflammasome components, and both signal 1 and in particular signal 2 are more efficient in untreated PWH.

### 2.4. Monocytes from PWH Are Not Primed for Inflammasome Activation In Vivo

To investigate whether monocytes in PWH with chronic HIV infection are primed for inflammasome activation in vivo, we activated monocytes from the same cohort with ATP without any LPS priming in vitro. There was no increase in IL-1α, IL-1β, or TNF release in monocytes from treated or untreated PWH compared to HIV-negative donor cells (Figure 4A–C). Furthermore, no difference in IL-1α and IL-1β levels were seen in plasma from PWH, treated or untreated, compared to HIV-negative donors (Figure 5A,B). These data could indicate that circulating monocytes in PWH are not significantly more primed for NLRP3 inflammasome activation as compared with circulating monocytes of HIV-negative donors, at least when ATP is used as signal 2.

## 3. Discussion

In this study, we investigated the expression of IFI16 and NLRP3 in circulating monocytes from ART-treated and untreated PWH. Our main findings can be summarized as follows: (i) IFI16 and NLRP3 expression correlated negatively with CD4 T cell counts, indicating that more progressive disease is related to upregulation of inflammasome proteins in monocytes; (ii) IFI16, but not NLRP3, was higher among PWH with high viral load; (iii) Monocytes from untreated PWH secreted more IL-1α and IL-1β as well as TNF than treated PWH upon stimulation; (iv) Monocytes from treated or untreated PWH were not pre-primed for inflammasome activation by ATP in vivo.

IFI16 and NLRP3 are important sensors in the innate immune response to HIV. Both IFI16 and NLRP3 are implicated in CD4 T cell depletion during HIV disease progression [24,25]. While IFI16 seem to induce pyroptosis in cells undergoing abortive infections, NLRP3 may induce pyroptosis in infected cells. Our findings herein suggest that both IFI16 and NLRP3 are regulated with higher expression in untreated PWH as compared with treated PWH. NLRP3 and IFI16 mRNA levels in monocytes were inversely correlated with CD4 T cell counts, and for IFI16 there was a gradual rise in relation to increasing viral load, further suggesting a link with HIV progression. 

Our findings may also be relevant in relation to HIV-related co-morbidities that are an increasing challenge even in the ART era. Thus, NLRP3 seems to link HIV infection to co-morbidities such as atherosclerotic cardiovascular disease [26] and neuroinflammation [27], both by inducing CD4 T cell depletion, a predictor of comorbidities [28], and by fueling chronic low-grade inflammation, in particular IL-1 activity, which has been linked to atherosclerotic events in PWH [22] and in the general population [29]. 

IFI16 recognizes lentiviral DNA in macrophages and CD4 T cells [23]. IFI16 is complex because it can both assemble inflammasomes and activate the IRF3 transcription factor, inducing interferon-β transcription [30,31]. This makes IFI16’s implication in the HIV pathogenesis logical and predictable. The role of NLRP3, on the other hand, is less obvious. However, NLRP3 is a remarkable cytosolic innate immune receptor, indirectly sensing the death of other cells without sensing a particular ligand [16]. Similarly, NLRP3 inflammasomes may be activated by pore forming toxins from microbes [11]. Phagocytes, such as macrophages and neutrophils, may also activate NLRP3 through phagocytosis of crystals made from urate or cholesterol, leading to similar intracellular distress due to “frustrated phagocytosis” [32]. Thus, at some stage during an HIV infection, IFI16-mediated pyroptosis may induce and activate NLRP3 in bystander cells, which in turn may contribute to chronic inflammation through additional release of IL-1α and IL-1β with or without pyroptosis. Recently, however, multiple inflammasome receptors, including NLRP3, were reported to synergize in pro-inflammatory cell death in bone marrow derived mouse macrophages, a process termed PANoptosis [33]. Although IFI16 was not investigated, this new concept of inflammasome activation leads to the hypothesis that IFI16 and NLRP3 may have simultaneous and converging actions during HIV induced CD4 T cell depletion.

In the present study, IL-1β and TNF release of monocytes from non-treated PWH was increased compared to monocytes from both treated PWH and HIV-negative donors (Figure 2B), which also correlated as expected with the pro-IL-1β mRNA expression (Figure 3B). TNF mRNA also featured an expected kinetic profile compared to its protein counterpart [34,35]. IL-1α release of monocytes from non-treated PWH was also increased compared to that of treated PWH (Figure 2A). However, we found no significant difference of NLRP3, IFI16, or IL-1α mRNA monocyte expression between these groups after LPS priming. Thus, the increased IL-1α cytokine release of monocytes from non-treated PWH does not seem to be explained by the levels of inflammasome receptor components or IL-1α synthesis. The regulation of IL-1α secretion is complex and may also be influenced by other factors not investigated in this study.

ATP did not induce IL-1α/IL-1β release from monocytes without LPS priming in vitro in any of the patient groups, including untreated PWH. Hence, our data do not suggest that circulating monocytes in PWH, whether ART-treated or not, are pre-primed for inflammasome activation. This is in contrast to our hypothesis that PWH would have already primed inflammasomes due to microbial translocation and low-grade immune activation [5]. However, our findings correspond to a recent report showing that although the NLRP3 inflammasome was upregulated in PWH with defective immune recovery, markers of microbial translocation were not elevated compared to immunological responders [19]. Notwithstanding, our findings also suggest that when exposed to both signal 1 and signal 2, monocytes from untreated PWH released increased levels of IL-1α and IL-1β, and this could clearly be relevant for the situation within the micro environment such as within the gastrointestinal tract where the cells are exposed to higher levels of LPS (signal 1) and within an atherosclerotic lesion where the cells are exposed to cholesterol crystals as signal 2 in NLRP3 activation [36]. Moreover, for PWH, in particular ART-naïve individuals [37], NLRP3 activation may be facilitated in vivo within the microenvironment due to a disturbed redox status [38].

A stronger activation to danger signals could be of relevance for vulnerability to certain risk factors for comorbidities, including cholesterol crystals, which are known to activate the NLRP3 inflammasome in the atherosclerotic process [31]. In our previous work, we showed that soluble markers of IL-1 activation measured at multiple time points over several years predicted first-time MI in PWH, but that the IL-1 activity remained mainly unchanged after ART initiation [39]. Of note, this increased risk was independent of HIV-related and traditional risk factors, including lipid profiles [22]. Moreover, in the present study, we also show increased release of alpha isoform of IL-1 from monocytes of untreated PWH, suggesting the involvement of gasdermin D. Notably, a preclinical study has suggested that IL-1α blockade affected early atherosclerosis, whereas anti-IL-1β treatment, but not IL-1α neutralization, limited progression and inflammation in established lesions [40]. These findings could also have implications for the management of atherosclerotic diseases in human, including PWH, as canakinumab, a monoclonal antibody against IL-1β, in contrast to anakinra, an IL-1 receptor blocker, do not inhibit IL-1α.

The present study has some limitations such as a relatively low number of individuals and lack of longitudinal data. The monocytes were isolated by plastic adherence. Although this method yields a purity of >90% monocytes [41], the cell culture is not completely void of lymphocytes which may affect the experiments. Moreover, lack of protein data for IFI16 and NLRP3 weaken our data on these molecules. Furthermore, because of lack of cell materials, we were not able to examine other relevant functions in monocytes such as gasdermin D-dependent cell death. Moreover, correlations and associations do not necessarily mean any causal relationship and we lack mechanistic data that could support our association data. Another major limitation is that IL-1α and IL-1β protein were detected with ELISA only and not by other methods such as Western blot. However, our findings further support the involvement of IFI16 and NLRP3 inflammasomes in HIV pathogenesis involving increased release of both IL-1 isotypes that may clearly also be related to the increased occurrence of certain comorbidities such as cardiovascular disorders in PWH. 

## 4. Material and Methods

### 4.1. Study Cohort

We included 20 untreated PWH, 22 ART-treated PWH, and 35 sex- and age-matched HIV-negative donors. Untreated PWH were recruited immediately prior to starting their ART treatment. Five participants were excluded due to experimental errors (2 HIV-negative donors, 3 untreated PWH) (Table 1). Of the remaining participants, RNA was collected from 54 participants (28 HIV-negative donors, 9 treated PWH, and 17 untreated PWH).

### 4.2. Reagents

Ultrapure LPS from Escherichia coli (O111:B4) was purchased from Invivogen, (Carlsbad, CA, USA). ATP, penicillin, and streptomycin were purchased from Sigma-Aldrich (St. Louis, MO, USA). RPMI 1640 medium containing stable glutamine and 25 mM HEPES (L0495) was obtained from Biowest (Nuaillé, France). Nunclon Delta surface cell culture dishes were purchased from Thermo Fisher Scientific (Waltham, MA, USA).

### 4.3. Cell Counts, Monocytes Isolation, Culture, and Stimulation

CD4 T-cells and HIV in peripheral blood were measured as part of clinical practice. For monocyte isolation, whole blood was collected into 8 mL sodium heparinized CPT vacutainer and inverted 10 times to ensure homogenization of the sodium heparin anti-coagulant and blood. The vacutainer was centrifuged at 1740× *g* for 20 min at room temperature, resulting in an upper layer of plasma over a cloudy band of PBMC. The PBMC layer and most of the plasma were pipetted into a 50 mL Falcon tube and the cells was spun down at 300× *g* for 10 min. Plasma was pipetted off and stored at −80 °C until analyses. The cell pellet was carefully resuspended and washed in RPMI 1640 twice (300× *g* for 10 min). The cells were then counted and seeded into 24 wells Nunclon Delta surface culture dishes at 300,000 cells/mL in RPMI 1640 without serum for 1 h for monocytes to adhere. The cells were then washed twice with RPMI 1640 to remove lymphocytes (purity of >90% monocytes [41]) and further cultured overnight in RPMI 1640 medium containing stable glutamine, 25 mM HEPES, 10% heat-inactivated FBS, 5 U penicillin/mL, and 50 μg/mL streptomycin. For NLRP3 inflammasome activation, monocytes were primed with LPS (0.1 ng/mL) for 1, 2, 4, and 6 h prior to harvesting, and then activated with 3 mM ATP for the last 30 min. 

### 4.4. ELISA

Conditioned media were collected and centrifuged at 300× *g* for 10 min to remove any detached cells, then stored at −80 °C until analysis. IL-1β and tumor necrosis factor (TNF) were quantified with DuoSet ELISA Kits (R&D Systems, Minneapolis, MN, USA). IL-1α were quantified with ELISA MAX^TM^ Deluxe Set Human IL-1α (sensitivity 0.6 pg/mL) (BioLegend, San Diego, CA, USA).

### 4.5. mRNA Quantification

RNA was extracted by using RNeasy Mini Kit (QIAGEN). cDNA was synthesized by using qScript cDNA SuperMix (Quantabio, Beverly, MA, USA). Real-time RT-PCR was performed with Brilliant III Ultra-Fast SYBR Green QPCR Master Mix (Agilent Technologies, Santa Clara, CA, USA) on a 7900HT Fast Real-Time PCR System (Thermo Fisher Scientific). Primer sequences are listed in Table 2. Relative gene expression was calculated using the ∆∆CT method.

### 4.6. Statistical Methods

Data were analyzed in Graphpad Prism ver. 6.0. Correlations were investigated with linear regression analysis. For comparisons of several means to a control mean, Dunnett’s test were used. For comparing selected independent means, Sidak’s multiple comparison test was used. For comparing every mean with every other mean, Tukey’s multiple comparisons test was used.

### 4.7. Ethics

The human samples (plasma/cDNA) were stored in biobanks approved by the Regional Committee for Medical Research Ethics South-Eastern Norway (permit numbers 2012/521, 2015/629 and 33256) and conducted according to the ethical guidelines outlined in the World Medical Association’s Declaration of Helsinki for use of human tissue and subjects. All participants gave written informed consent.

## Figures and Tables

**Figure 1 ijms-25-07141-f001:**
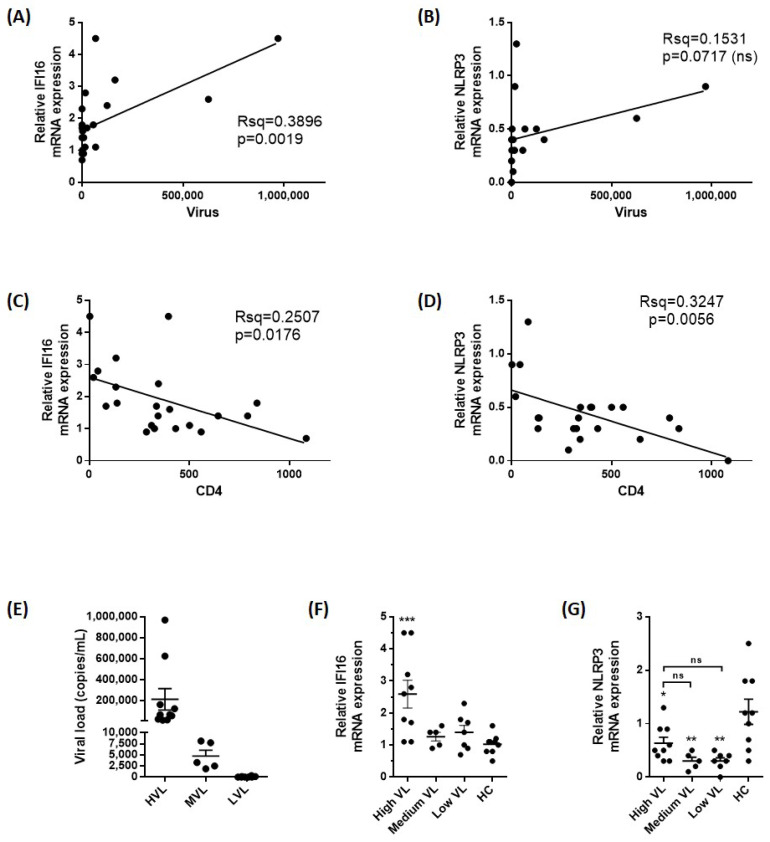
The correlation of inflammasome expression in monocytes with viral load and CD4 numbers in PWH (n = 21). (**A**) Linear regression analysis reveals that IFI16 mRNA expression in monocytes increases with higher viral load. (**B**) A similar correlation pattern with NLRP3 mRNA expression was not significant. (**C**,**D**) IFI16 and NLRP3 mRNA expression in monocytes decrease with increasing CD4 counts (linear regression analysis). (**E**) PWH were stratified according to viral load: high viral load (HVL, >10,000 copies/mL, n = 9), medium viral load (MVL, 1000–9999 copies/mL, n = 5), and low viral load (<1000 copies/mL, n = 7). Lines are mean viral load in blood (copies/mL) with standard error of the mean (SEM). (**F**,**G**) IFI16 and NLRP3 mRNA expression were quantified in the stratified patient groups presented in (**E**) and compared to gender- and age-matched HIV-negative donors (HC, n = 9). Mean with SEM are indicated in the column dotplots. *** *p* < 0.001, ** *p* < 0.01, * *p* < 0.05 compared to HIV-negative donors (HC) (**F**,**G**), (Dunnett’s multiple comparisons test) or as indicated (Sidak’s multiple comparisons test (**G**)). ns: not significant.

**Figure 2 ijms-25-07141-f002:**
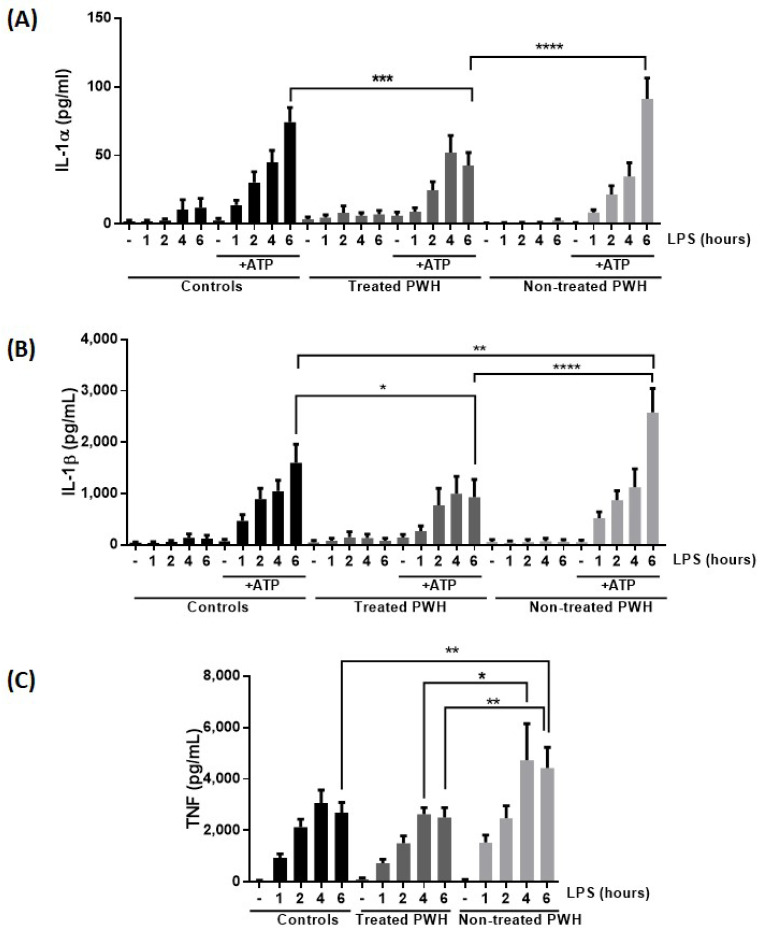
IL-1α, IL-1β, and TNF release from monocytes are increased in non-treated PWH compared to treated PWH. (**A**,**B**) Monocytes were isolated from peripheral blood by plastic adherence. The cells were primed with medium only or 0.1 ng/mL LPS for 1–6 h as indicated and then exposed to the inflammasome activator ATP (3 mM) for 30 min. IL-1α and IL-1β were quantified in the conditioned medium with ELISA. The corresponding timepoints with peak secretion were compared **** *p* < 0.0001, *** *p* < 0.001, ** *p* < 0.01, * *p* < 0.05 (Sidak’s multiple comparisons test). (**C**) TNF release from the LPS-primed cells in was quantified in the same conditioned media. The corresponding time points with peak secretion were compared ** *p* < 0.01, * *p* < 0.05 (Sidak’s multiple comparisons test). HIV-negative donors (controls) (n = 33), treated PWH (n = 21), and untreated PWH (n = 17).

**Figure 3 ijms-25-07141-f003:**
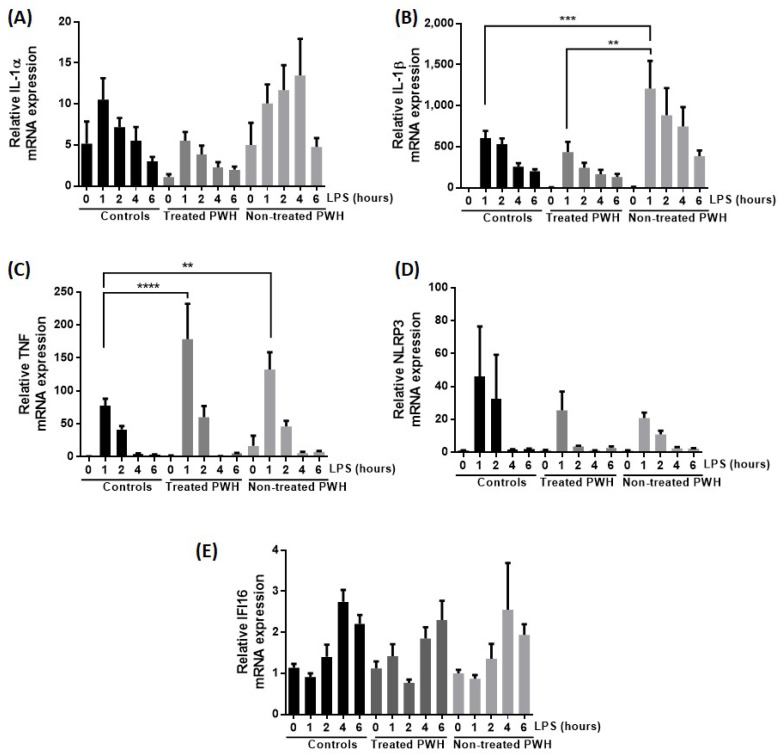
LPS-induced IL-1α, IL-1β, TNF, NLRP3, and IFI16 mRNA expression in monocytes from PWH with HIV compared to HIV-negative donors. Cells were incubated with medium only or 0.1 ng/mL LPS for 1–6 h as indicated. mRNA was isolated and the indicated mRNA expression quantified with RT-PCR. (**A**) IL-1α. (**B**) IL-1β. (**C**) TNF. (**D**) NLRP3. (**E**) IFI16. Columns are mean with SEM. The corresponding timepoints with peak expression were compared. **** *p* < 0.0001, *** *p* < 0.001, ** *p* < 0.01, (Sidak’s multiple comparisons test). HIV-negative donors (controls) (n = 33), treated PWH (n = 21), and untreated PWH (n = 17).

**Figure 4 ijms-25-07141-f004:**
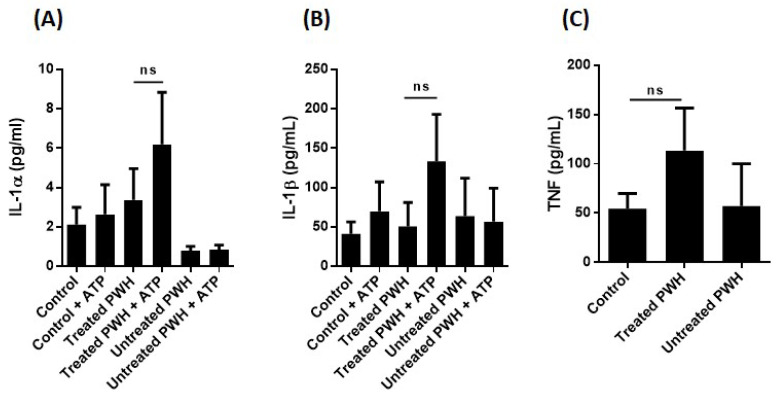
No sign of in vivo priming of monocytes from PWH. (**A**,**B**) Monocytes were incubated with medium only for 6.5 h or exposed to 3 mM ATP for the last 30 min as indicated. IL-1α and IL-1β were quantified in the conditioned media and the levels of ATP-treated cells compared to corresponding untreated control cells (Sidak’s multiple comparisons test). There are no statistically significant differences (ns). (**C**) TNF was quantified in the same conditioned media as indicated and all mean levels were compared to (Tukey’s multiple comparisons test). There are no statistically significant differences. HIV-negative donors (controls) (n = 33), treated PWH (n = 21), and untreated PWH (n = 17).

**Figure 5 ijms-25-07141-f005:**
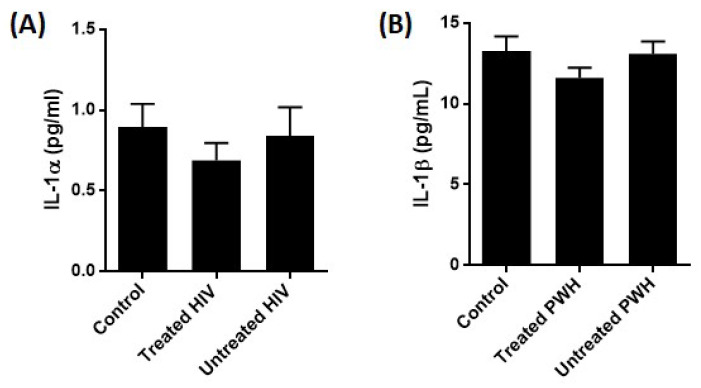
Inflammasome-dependent cytokines in plasma from PWH and HIV-negative donors controls. (**A**,**B**) Plasma were obtained from HIV-negative donors (n = 33), treated PWH (n = 21), and untreated PWH (n = 17). IL-1α and IL-1β were quantified with ELISA. Columns are mean with SEM. Means were compared with Tukey’s multiple comparison test, there are no statistically significant differences.

**Table 1 ijms-25-07141-t001:** PWH and HIV-negative control demographics.

	HIV-Negative Donors	Treated PWH	Untreated PWH
n	33	22	17
Sex, male, (%)	20 (61)	12 (57)	14 (82)
Age, years, median (IQR)	47 (41, 56)	51 (44, 56)	38 (31, 56)
Race, n (%)			
White	29 (88)	19 (86)	13 (76)
Black	0 (0)	3 (14)	1 (6)
Asian	4 (12)	0 (0)	3 (18)
Years from diagnosis, median (IQR)		20 (14, 28)	0
Months from diagnosis to ART, median (IQR)		69 (40, 171)	
Nadir CD4, cells/μL, median (IQR)		180 (130, 228)	398 (202, 552)
CD4, cells/μL, median (IQR)		628 (448, 786)	410 (202, 552)
CD8, cells/μL, median (IQR)		780 (618, 1029)	875 (682, 1103)
Viral load, copies/mL, median (IQR)		0	75,000 (47,000, 295,000)
Participants included for PCR analysis, n	28	9	17
IQR: interquartile range			

**Table 2 ijms-25-07141-t002:** Primer sequences.

Target mRNA	Primer
IL-1α	Forward: 5′ ATCCTGGAAGGAGGAAGGAA-3′ Reverse: 5′ GCTGTTCATGGTCAGGGAAT-3′
IL-1β	Forward: 5′-CCCTAAACAGATGAAGTGCTCCTT-3′ Reverse: 5′-GGTGGTCGGAGATTCGTAGCT-3′
TNF	Forward: 5′-CCAGGCAGTCAGATCATCTTCTC-3′ Reverse: 5′-GGAGCTGCCCCTCAGCTT-3′
NLRP3	Forward: 5′-AGCTTCAGGTGTTGGAATTAGACA-3′ Reverse: 5′-GCTGGAGGTCAGAAGTGTGGA-3′
IFI16	Forward: 5′ CCCAACAGTTCTTCAACTGAGAAC-3′ Reverse: 5′ GTCTCATATTCAAATGGCTTTGTTTGTA-3′
TATA Binding Protein	Forward: 5′ AGTGAACATCATGGATCAGAACAAC-3′ Reverse: 5′ AGATAGGGATTCCGGGAGTCA-3′

## Data Availability

The raw data supporting the conclusions of this article can be made available by the authors on request.

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
