# Peer review of "Chronic HIV Infection Increases Monocyte NLRP3 Inflammasome-Dependent IL-1α and IL-1β Release"

_ijms, 2024, doi:10.3390/ijms25137141_

Round 1

Reviewer 1 Report

Comments and Suggestions for Authors

In this study, Hoel et al., explores IFI16 and NLRP3 inflammasome expression and function in monocytes from ART treated or untreated PLWH and healthy people, and found that IFI16 mRNA expression in unstimulated monocytes correlated positively with viremia, and correlated negatively with CD4 lymphocyte levels in untreated PLWH. The study was interesting, but some of the results were arbitrary and not supported by clear experimental results.

 There are some major concerns:

1. In lines 108 to 110, the conclusion “both IFI16 and NLRP3 expression is likewise higher in patients with more pronounced immunodeficiency” seems arbitrary. Figures 1B and 1G showed that NLRP3 expression is not significantly associated with viral load. This issue should be clarified in the whole text.

 2. For Figure 2, it is necessary to detect IL-1 production by WB. In addition, the expression of NLRP3 and IFI16 also needs to be examined.

 3. Figure 2C and Figure 3C, it is difficult to understand why the expression of TNF at the protein level and mRNA level is inconsistent.

Reviewer 2 Report

Comments and Suggestions for Authors

Comments on the Quality of English Language

The manuscript needs an extensive review of the grammar and terminology used. PLWH is no longer used to define people with HIV and has been replaced by PWH, it’s also suggested the use of HIV- donors instead of healthy donors. I would also recommend using PWH instead of patients.

Reviewer 3 Report

Comments and Suggestions for Authors

In the manuscript, Hoel and colleagues compare IFI16 and NLRP3 mRNA expressions in monocytes from healthy controls and HIV-infected individuals, with or without antiretroviral therapy (ART), demonstrating that both gene expressions are negatively correlated with CD4 counts and partially positively correlated with viral load. The authors also show that monocytes from untreated HIV-infected individuals exhibit increased release of IL-1α, IL-1β, and TNF compared to those from treated ones.

The results are convincing and generally support the conclusion. IFI16 and NLRP3 expression might be good indicator for the prediction of disease progression. Overall, this is a well-written manuscript with some minor deficiency outlined below. The strength of this manuscript can be significantly improved by addressing them.

1. For Fig 1A, 1B and 1E, viral load is typically analyzed in logarithmic units (log) rather than in raw counts, since viral load can vary greatly even within the same individual over time.

2. Please verify it’s inflammasome expression of PBMCs or monocyte in Fig1 legend.

3. Author didn’t use isolation kit to purify monocytes from PBMCs. Can author show the purity of monocytes after enrichment?

4. It's very interesting to see that 6-hour LPS treatment induces the highest levels of IL1 release but not cytosol mRNA levels. Does LPS treatment also increase the expression level of TATA-binding protein, which affects the normalization of gene expression?"

5. The results from Figure 3 show that there were no statistically significant differences in IL-1α, NLRP3, or IFI16 mRNA expression between the groups. How does the author explain that it is NLRP3-dependent, as the title claims?

6. What is the detection limit of the IL-1α ELISA assay? I think that the results of Figures 4A and 5A might be considered undetectable.

7. The author cited previous work showing that NLRP3 and IFI16-mediated inflammation can cause GASMD-dependent cell death. Did the author observe more cell death in monocytes from HIV-infected individuals than in those from healthy controls after LPS treatment?

Round 2

Reviewer 1 Report

Comments and Suggestions for Authors

The authors did not address the reviewer's comments

Comments on the Quality of English Language

No

Reviewer 2 Report

Comments and Suggestions for Authors

I accept all the changes made by the authors

Round 3

Reviewer 1 Report

Comments and Suggestions for Authors

In Figure 2, the authors need to isolate new PBMC to detect the expression of NLRP3 and IFI16 
